# Therapeutic Potential of Human Immature Dental Pulp Stem Cells Observed in Mouse Model for Acquired Aplastic Anemia

**DOI:** 10.3390/cells11142252

**Published:** 2022-07-21

**Authors:** Vivian Fonseca Gonzaga, Cristiane Valverde Wenceslau, Daniel Perez Vieira, Bruna de Oliveira Policiquio, Charbel Khalil, Rodrigo Pinheiro Araldi, Irina Kerkis

**Affiliations:** 1Genetics Laboratory, Instituto Butantan, São Paulo 05503-900, Brazil; vivia_gonzaga@hotmail.com (V.F.G.); bruna.policiquio@gmail.com (B.d.O.P.); rodrigo.pinheiro.araldi@gmail.com (R.P.A.); 2Cellavita Pesquisas Científicas Ltda., Valinhos 13271-650, Brazil; cristiane.valverde@cellavitabrasil.com.br; 3Nuclear and Energy Research Institute, São Paulo 05508-000, Brazil; dpvieira@ipen.br; 4Reviva Stem Cell Platform for Research and Applications Center, Bsalim 17-5208, Lebanon; charbelk3@hotmail.com; 5Faculty of Pharmacy, Saint Joseph University, Beirut 17-5208, Lebanon

**Keywords:** aplastic anemia (AA), human immature dental pulp stem cells (hIDPSC), mesenchymal stem cells (MSC), total body irradiation (TBI), hematopoiesis recovery

## Abstract

Aplastic anemia (AA) is a rare and serious disorder of hematopoietic stem cells (HSCs) that results in the loss of blood cells due to the failure of the bone marrow (BM). Although BM transplantation is used to treat AA, its use is limited by donor availability. In this sense, mesenchymal stem cells (MSCs) can offer a novel therapeutic approach for AA. This is because the MSCs contribute to the hematopoietic niche organization through their repopulating. In our study, we used the human immature dental pulp stem cell (hIDPSC), an MSC-like cell, to explore an alternative therapeutic approach for AA. For this, isogenic C57BL/6 mice were exposed to total body irradiation (TBI) to induce the AA. After 48 h of TBI, the mice were intraperitoneally treated with hIDPSC. The immunohistochemistry analyses confirmed that the hIDPSCs migrated and grafted in the mouse bone marrow (BM) and spleen, providing rapid support to hematopoiesis recovery compared to the group exposed to radiation, but not to those treated with the cells as well as the hematological parameters. Six months after the last hIDPSC transplantation, the BM showed long-term stable hematopoiesis. Our data highlight the therapeutic plasticity and hematoprotective role of hIDPSC for AA and potentially for other hematopoietic failures.

## 1. Introduction

Aplastic anemia (AA) is a rare, life-threatening bone marrow (BM) failure syndrome, characterized by hypoplastic and fatty BM, with profound reductions in the hematopoietic stem/progenitor cells (HSPCs) that lead to defective mature blood cell production and peripheral pancytopenia [1]. AA can be inherited or acquired; the acquired form is the most prevalent, with a higher incidence reported in Asia and Europe. Regardless of etiology, all patients present anemia symptoms, bleeding, and infections. Although the pathophysiology of acquired AA remains unknown, cumulative evidence has supported the understanding that the immune-mediated destruction of HSPCs plays a central role in the pathogenesis of acquired AA [2]. In this sense, several environmental factors, including drugs, chemicals, radiation, and viruses have already been identified as responsible for dysregulating CD8+ cytotoxic T cells and CD4+ T cells, including T helper type 1 (Th1) and 2 (Th2), regulatory T (Treg) and Th17 cells, and natural killer (NK), leading to the abnormal production of cytokines, such as tumor necrosis factor (TNF)-α and transforming growth factor (TGF)-β, which induce apoptosis of HSPCs [2,3]. 

The stem cells in the BM are crucial to maintaining blood cell production [4,5,6]. These cells reside within the hematopoietic stem cell (HSC) niche, which consists of a variety of HSC and non-hematopoietic cells [7], including bone marrow mesenchymal stromal/stem cells (BM-MSCs), which are mandatory for the support of long-term hematopoiesis [8]. Due to their capability to provide microenvironmental support for the HSCs [9] and differentiate into various mesodermal lineages [4,5,6], the mesenchymal stem cells (MSCs) have been explored as candidates for the treatment of (AA) [10,11,12], as an alternative to the treatment with immune system-suppressing therapy or HSC (BM) transplantation [5,9]. This is because the MSCs can elicit reparative and immunomodulatory effects through paracrine and systemic mechanisms, benefiting many organs [13,14,15,16] in a manner that is independent of the long-term engraft [17]. Although the dental pulp stem cells (DPSCs) differ from BM-MSCs in both embryonic origin (neural crest) and biological role, these cells have a high regenerative and immunomodulatory capacity [18,19,20]. Moreover, these cells represent an alternative source of stem cells as they can be easily isolated from human deciduous and adult teeth and wisdom teeth [21,22]. Despite the ectomesenchyme origin of these cells, the DPSCs exhibit all the criteria for defining MSCs, as defined by the International Society of Cellular Therapy [22] (ISCT). These characteristics can be related to the biological function of these cells, which are responsible for the maintenance and repair of the periodontal tissue [23], supporting the DPSC application in dentistry for the regeneration of tooth structures [24,25,26] and neurogenesis [27,28]. Based on this, we investigated here, for the first time, the therapeutic potential of human immature dental pulp stem cells (hIDPSCs) for the treatment of AA in a mouse model for AA because the safety and therapeutic potential and multifunctionality of these cells have previously been demonstrated [20,21,29,30,31,32,33]. Although different mouse models of acquired BM failure have been used to understand the physiopathology of AA or have been employed in preclinical studies, the BM failure induced by radiation is the most commonly used [11,32,34,35,36,37]. Radiation is an effective tool, even at low exposures, to induce hematopoietic failure. However, high irradiation doses provoke acute radiation syndrome. The latter manifests as severe neutropenia and thrombocytopenia, leading to infection, bleeding, and death [38,39]. Therefore, this model is widely used in preclinical studies to evaluate the potential of the MSC effect on AA. Currently, clinical and preclinical studies provide evidence showing that BM MSC transplantation can prevent hematopoietic graft failure when administered alone or cooperatively with hematopoietic cells [10,40,41,42,43,44]. 

## 2. Materials and Methods

### 2.1. Ethical Aspects

All the procedures involving the animals were approved by the Ethics Committee on the Use of Animals of the Butantan Institute (CEUA/IB, process number 01201/14). The protocols concerning the experimental animals’ maintenance, care, and handling are all in accordance with the current Brazilian legislation and internationally recognized norms and protocols. All staff members working with the experimental animals were fully accredited as staff researchers/technicians and were adequately trained in the use of animals for experimental scientific purposes according to the current Brazilian regulations.

### 2.2. Cell Culture and Characterization

The hIDPSC used in this study were isolated from the dental pulp (ectomesenchymal origin) of the deciduous teeth of a 6-year-old individual (male) and fully characterized according to the protocol developed by Kerkis et al. [20] and patented with the US patent number 9790468B2 (available in https://patents.google.com/patent/US9790468B2/en accessed on 16 May 2022). The hIDPSC isolation technology was licensed by the Brazilian facility Cellavita Pesquisas Científicas Ltd., which produced these cells by a current, good manufacturing process (c-GMP). The hIDPSCs produced by Cellavita are cultivated until the fifth passage (P5) in Dulbecco’s modified Eagle’s medium (DMEM)/Ham’s F12 (1:1; Invitrogen, Waltham, CA, USA), supplemented with 15% fetal bovine serum (FBS; HyClone, Logan, UT, USA), 2 mM glutamine (Gibco, Gaithersburg, MD, USA), 50 mg/mL gentamicin sulfate (Schering-Plough, Whitehouse Station, NJ, USA), and 1% nonessential amino acid (Gibco, Carlsbad, CA, USA) at 37 °C in a 5% CO_2_ high-humidity atmosphere. The hIDPSCs in P5 correspond to the active component of the NestaCell^®^ product (Cellavita Pesquisas Científicas Ltda., Valinhos-SP, Brazil), whose safety and therapeutic potential were already investigated in phase I and II clinical trials for the treatment of Huntington’s disease (ClinicalTrial.gov identifier NCT02728115 and NCT03252535, respectively) and severe COVID-19 pneumonia (NCT04315987). 

As previously reported, the hIDPSCs employed in this study are considered as MSC-like cells, which expresses the typical MSC markers proposed by the International Society for Cellular and Gene Therapy (ISCGT) [1], being positive for CD105, CD73, CD90, and CD44 and negative for CD45, CD34, CD14, and HLA class II [2,3,4]. However, unlike other typical MSCs, including stem cells from exfoliated deciduous teeth (SHED), the hIDPSCs express and secrete high levels of the neuronal marker nestin, as previously demonstrated by Kerkis and Caplan [21].

### 2.3. Developing Mice Model for AA

To evaluate the therapeutic efficiency of hIDPSCs for AA, 62 isogenic C57BL/6 female mice (n = 62), aged 4–6 weeks, were used in this study. The animals were obtained from the Central Laboratory of Animals (Butantan Institute, São Paulo, Brazil). They were maintained in the bioterium of the Laboratory of Genetics (Butantan Institute) in a 12:12 h light/dark cycle, with a constant temperature of 22 ± 1 °C, receiving water and food ad libitum. A total of 56 mice (n = 56) were exposed for 31 min to a panoramic radiation source containing Cobalt 60. The mice were positioned 40 cm away from the radiation source and received a standardized radiation dose of 6 Gy, as proposed by Vieira et al. [23]. The total body irradiation (TBI) was performed in the Nuclear and Energy Research Institute (IPEN) of the University of São Paulo (USP, Brazil). As a control, six mice (n = 6) were used from the irradiated group of 56; they were euthanized 48 h after TBI radiation to confirm the BM ablation (positive control). In addition, six non-irradiated mice (n = 6) were used; they were employed as a negative control. 

### 2.4. Cell Therapy

After 48 h of TBI exposure, six mice (n = 6, positive control) were euthanized to confirm the BM ablation. The other 50 mice (n = 50) subjected to TBI exposure were divided into two groups. The first group (n = 25) was intraperitoneally treated with three doses of 1 × 10^6^ hIDPSC/animal. The first cell dose was administrated 48 h after the TBI exposure (day 2, D2); the second cell dose was given 15 days after the first cell dose (day 17, D17); and the last cell dose was given 15 days after the second cell dose (day 32, D32). The second group (n = 25) was intraperitoneally treated with three doses of 100 μL of saline (0.9% NaCl solution, placebo), which was used as a vehicle for the hIDPSCs. The saline was administered using the same posology employed in the mouse group which had received the cells. The first saline administration occurred 48 h after the TBI exposure, on D2; the second administration was on D17; and the third saline administration was on D32. Forty mice (n = 40) were euthanized 30 days after the third cell (n = 20) or saline (n = 20) administration (short-term/D62), while ten animals (n = 10) were euthanized only 6 months after the third cell (n = 5) or saline (n = 5) administration (long-term/D182). A schematic cell therapy schedule is shown in Figure 1A,B.

### 2.5. Immunofluorescence (IF) Analyses

The BM cells were collected and fixed in 4% paraformaldehyde solution (in PBS) for 20 min and washed with TBS (150 mM NaCl, 50 mM Tris-HCl pH 7.6) twice. The cells were permeabilized with 0.1% Triton X-100 Sigma (Sigma-Aldrich, St. Louis, MO, USA) for 20 min and incubated for 30 min with a 5% BSA solution in PBS (Sigma-Aldrich). The cells were incubated overnight at 4 °C with the primary antibodies described in Table 1. The cells were washed 1× with PBS and then incubated for 20 min at room temperature with the appropriate secondary antibody (Table 2). The nucleus was stained with DAPI (Vector Laboratories, Ltd., Burlingame, CA, USA). The material was analyzed using the NIKON DS-Ri1 fluorescence microscope (with HBO lamp excitation light for DAPI fluorochrome, excitation laser 488 nm for FITC) (Nikon, Tokyo, Japan).

### 2.6. Histological and Immunohistochemistry (IHC-P) Analyses

The femurs and spleen collected on D32 or D182 (as described in Figure 1) were fixed in 10% paraformaldehyde solution for 24 h. Then, the femurs were decalcified in descaler solution (0.5 M EDTA and 10% formaldehyde) for seven days, as proposed by Junqueira and Carneiro 2004 [22]. The tissue samples (femurs and spleen) were embedded in paraffin and sectioned in 5 μm sections. The femur and spleen sections were stained with hematoxylin and eosin (HE) (Merck, Darmstadt, Germany) for histological analysis. Immunohistochemistry (HC-P) analyses were performed using the femur and spleen samples collected. For this analysis, the sections were deparaffinized and subjected to antigen retrieval in 0.01 M citrate buffer at 95 °C for 35 min. The IHC-P was performed using the EnVisionTM + HRP System (Dako, Santa Clara, CA, USA). Endogenous peroxidase was blocked with the peroxidase block included in the kit for 10 min. The sections were washed three times in tris-buffered saline (TBS) (50 mM Tris-Cl, 150 mM NaCl, pH 7.6) for 5 min and permeabilized with 0.01% Triton X-100 for 20 min. After the permeabilization, the sections were washed for 5 min in TBS and blocked with 5% BSA solution (in PBS) for 40 min. The sections were incubated overnight at 4 °C in a humid chamber with the primary antibodies described in Table 1. After the incubation, the slides were washed in TBS-T (TBS, 0.2% Tween-20) and incubated in a humid chamber for 1 h at room temperature with the secondary antibodies described in Table 2. The sections were washed three times in TBS-T for 5 min and treated with two drops of chromogenic substrate AEC for 15 min. The material was washed in distilled water for 5 min and counter-stained with Mayer-hematoxylin for 3 min. The slides were rinsed with distilled water and dipped in a 0.037 M ammonium hydroxide solution to remove the hematoxylin excess. The slides were mounted with Paramount mounting medium (Dako, Santa Clara, CA, USA). The slides were analyzed using a binocular light microscope Axiophot (Carl Zeiss, Jena, Germany) with objectives of 10×, 20×, and 40×. The images were captured using the AxioVision software version 4.7.2. (Carl Zeiss, Jena, Germany).

### 2.7. Flow Cytometry (FC) Analysis

The BM cells collected from the mouse femurs were previously fixed to 0.1% paraformaldehyde for 20 min and washed with TBS. The cells were permeabilized with 0.1% Triton X-100 (Sigma-Aldrich, St. Louis, MO, USA) and incubated for 30 min with a 5% BSA solution. Next, the cells were incubated for 20 min at 4 °C with the same primary used for IF (Table 1). After the incubation, the cells were washed in TBS and incubated for 20 min at room temperature with the same secondary antibodies used for IF (Table 2). The acquisition was performed using the flow cytometer BD Accuri^®^ C6 (BD Bioscience, Bogota, NJ, USA). For this analysis, a total of 10,000 events were acquired. The results were analyzed using FlowJo v10 software version (TreeStar, Woodburn, OR, USA).

### 2.8. BM Colony-Forming Unit Assay (CFU)

Aiming to evaluate whether the hIDPSC transplantation could protect the endogenous BM stem cells against the radiation-induced damage, we performed the CFU assay. The BM was aspirated from the mouse femurs, according to Wenceslau et al. [45]. The aspirate was subjected to cell counting using a Neubauer chamber. A total of 4 × 10^5^ cells from the BM aspirate were seeded in a 33 mm-diameter cell culture dish (Techno Plastic Products (TPP), Trasadingen, Switzerland) to evaluate the BM-CFU capacity. The cells were cultivated for 21 days using DMEM-Low Glucose, supplemented with 10% fetal bovine serum, 1% non-essential amino acids, 1% L-glutamine, and penicillin (100 U/mL/streptomycin (100 μg/mL) solution (all from Gibco, Carlsbad, CA, USA). The cells were maintained in a humidified atmosphere of a 5% CO_2_ atmosphere at 37 °C. After 21 days, the cells were fixed in 4% paraformaldehyde solution for 24 h and stained with 1% violet crystal (Millipore, Burlington, MA, USA). Cell clusters consisting of at least 50 cells were classified as a BM-CFU and counted.

### 2.9. Blood Cell Count

To evaluate the therapeutic effects of hIDPSCs, the blood was collected via retroorbital puncture using the BD Vacutainer tube with K2 EDTA (BD Bioscience, Bogota, NJ, USA). The samples were collected before TBI (D0), after 48 h of TBI exposure (D2), and at each cell or saline administration (D2, D17, and D32). Blood was also collected during the euthanasia (D62/D182). The total cell count (i.e., white blood cell (WBC), neutrophil, lymphocyte, monocyte, eosinophil, basophil, red blood cell (RBC), and hematocrit) was performed using the veterinary hematology analyzer BC-2800Vet^®^ (Mindray, Shenzhen, China). In addition, morphological analyses were obtained through blood smears, which were stained with NewProv PanPro Instant Provider (NewProv, Parana, Brazil) and analyzed using the Nikon DS-Ri1 microscope (Nikon, Tokyo, Japan). Furthermore, 30 days (D62) and 6 months (D182) after the last hIDPSC transplantation (D32), the spleen, BM aspirate, and femur were collected for morphological and immunohistochemical analysis (Figure 1).

### 2.10. Statistical Analysis

Statistical analyses were performed using the one-way analysis of variance (ANOVA), followed by the Tukey test, both with a significance level of 5%, using the GraphPad Prism 5.02 software. 

## 3. Results

### 3.1. Immunophenotyping of hIDPSC

Although the expression of the conventional MSC markers [22] and the hIDPSC marker (nestin) was already extensively demonstrated in the hIDPSCs [20,21,30], we analyzed the expression of the MSC biomarker involvement in hematopoiesis and the protective or reparative processes [10,45,46], such as CD44, CD90, fibronectin, and vimentin [47]. The results confirmed that the hIDPSCs used in this study express these biomarkers (Figure 2). We also investigated the expression of nestin, a marker of immature neural cells, which is highly expressed by the hIDPSCs and is recognized as a differential marker of hIDPSCs and SHED cells, as previously reported by us [20,35,48]. The results showed that the hIDPSCs express nestin (Figure 2B), reinforcing the therapeutic potential of these cells for the AA treatment because nestin is naturally expressed by a subset of the BM perivascular MSCs, which contact closely with HSPCs [21]. 

### 3.2. Short- and Long-Term Benefits Observed in BM of an Irradiated Mouse after hIDPSC Transplantation

First, we performed a histopathological analysis of mouse BM after irradiation. The femurs collected 48 h after the TBI showed apparent tissue damage and BM destruction (Figure 3C,D). The BM destruction was evidenced by the disseminated lesion, along with the medullary tissue and the reduction in cellularity (hypoplasia/medullary aplasia), the absence of medullary precursor cells in general, the infiltration of red blood cells, the more significant amount of spinal stromal, and the substitution of BM by fat elements in some regions (Figure 3D). By contrast, the non-irradiated mice did not present any morphological change in the BM (Figure 3A,B). These results confirm that the irradiation procedure was efficient in inducing AA in mice. In addition, the irradiated and hIDPSC-transplanted mice showed increased cellularity in the BM (Figure 3E,F) compared with those that were irradiated and treated with saline (Figure 3G,H). We identified the presence of the spinal canal filled with medullary precursor cells in general, such as megakaryocytes into the BM of the hIDPSC-transplanted mice (Appendix A). Interestingly, the increased cellularity in the BM of the irradiated and hIDPSC-treated mice was observed six months after the hIDPSC transplantation (Figure 3I,J). By contrast, the irradiated and saline-treated mice showed an expressive hypocellularity accompanied by fat tissue replacement (Figure 3K,L). These data suggest that short- (D62) and long-term (D180) treatment could potentially recover the BM of the mouse model for AA.

### 3.3. Long-Term hIDPSC Homing in Irradiated BM

To analyze the long-term homing of the hIDPSCs, 30 days after the third hIDPSC transplantation (D62) the mice were euthanized, and the BM and spleen were collected and subjected to the immunodetection using the anti-human nucleus (hNu) antibody by IHC-P. The results showed hNu+ cells in the BM in the irradiated and the hIDPSC-treated mice (Figure 4A–C). However, in the irradiated and non-treated mouse BM, we did not observe any presence of hNu+ cells (Figure 4D). Similar results were observed in the spleen of the irradiated and hIDPSC-treated mice (Figure 4E–G) and the irradiated and non-treated mouse spleen (Figure 4H). These data support the idea that the human cells can show long-term survival in the mouse BM and spleen, the main hematopoietic organs.

### 3.4. Evaluation of Expression of Endogenous Stromal and Hematopoietic Cell Markers after hIDPSC Transplantation

To evaluate the therapeutic potential of hIDPSCs, we analyzed the percentage of nestin- and CD44-positive cells. This is because cells positive for these biomarkers regulate HSC and, therefore, the BM microenvironment [49,50,51]. We observed that the exposure to TBI statistically reduced the number of nestin-positive cells, as expected after irradiation. However, only the group treated with hIDPSCs showed a significant increase in nestin-positive cells, while the saline group did not show a statistical increase. Moreover, the experimental groups presented a significant difference, where the nestin-positive cells of the group treated with hIDPSCs were statistically higher than the saline group (Figure 5A). The exposure to TBI also provoked a statistical reduction in the number of CD44-positive cells. Even though both experimental groups showed a significant increase in CD44-positive cells in relation to the irradiated, this increase was higher in those treated with hIDPSCs (**) than the saline group (*) but did not show a significant difference between them (Figure 5B). The results showed that after 6 months or in the long term (D182), the saline group no longer showed a statistical difference in comparison with the hIDPSC-treated group, as previously demonstrated in the short term (D62) in the nestin-positive cells, while we observed that the CD44-positive cells were the reverse. Only after 6 months or in the long term (D182), the saline group showed a statistical difference in comparison with the hIDPSC-treated group, whereas previously no short-term statistical difference was shown between these experimental groups (Figure 5C).

These results may not confirm that the expression of nestin and CD44 comes particularly from the hIDPSC engraftment in the medullary environment or even if the treatment was responsible for the observed increase; such markers can also be found in the basal BM, and the saline group also mostly showed an increase in relation to the irradiated group. Despite this, we suggest that the hIDPSC treatment may led to an increase, mainly of the nestin-positive cells in the short term, whereas hIDPSC treatment may cause an increase in CD44-positive cells only 6 months later (long-term). 

### 3.5. CFU-f Assay

The cells aspirated from the mouse BM were subjected to the CFU assay to verify if the hIDPSC transplantation could protect the intrinsic BM cells from the radiation-induced damages 48 h after the TBI exposure. The results showed a significant decrease in cell colonies after TBI exposure (Figure 6). By contrast, the hIDPSC-treated group showed a significant increase in cell colonies, while the saline group did not show a statistical increase. Furthermore, we observed a statistical difference between the hIDPSC-treated group and the saline group (Figure 6). These data suggest that hIDPSC transplantation can confer a protective effect for BM cells. 

### 3.6. Monitoring of Hematological Parameters

The hematological parameters (red blood cells—RBCs and white blood cells—WBCs) were evaluated to investigate the influence of the hIDPSC treatment on the blood cell recovery. After the irradiation, we observed a significant decrease in the RBC number of the hIDPSC-treated group on D2 until D17 and WBC on D2, D17, D32, and D62 until D182 in relation to D0 (basal control) (Figure 7A,C). However, only the treatment with hIDPSCs statistically increased the number of RBCs and WBCs after the second cell infusion (D32, Figure 7), but not after the first cell infusion (D17, Figure 7). These results suggest that at least two hIDPSC doses and an interval of 30 days are necessary to verify the functional effects of the cell therapy on the hematological parameters. 

After the significant increase presented on D32, the RBC values of the hIDPSC-treated group did not show a statistical decrease until D182 (Figure 7A). In contrast, we observed a reduction in the WBC number of the hIDPSC-treated group on D62 and D182 (Figure 7C), suggesting that the benefits of cell therapy with hIDPSCs are most prominent for the RBCs.

On the other hand, after the irradiation, the saline group only presented several consecutive decreases until D62 in the RBC values and until D182 in the WBC values and did not show an increase at any point (Figure 7B–D). These results demonstrated that the saline group was not able to naturally increase the RBCs and WBCs at any point of the study. 

## 4. Discussion

The hematopoietic stem cell (HSC) is a major therapeutic option for patients with acquired severe aplastic anemia. However, the difficulty in finding hemocompatible donors or the failures related to the HSC engraftment limit the clinical potential of HSC-based therapy. For this reason, novel therapeutic approaches that can reduce these limitations are mandatory to improve the outcome of patients with AA. Previous studies used MSC derived from different sources originated from mesoderm or extraembryonic tissue, such as BM, umbilical cord, adipose tissue, and placenta, in the therapeutic field as well as for hematologic disorders [10]. In this sense, the current studies have shown that nestin-positive mesenchymal stem cells (MSCs) are not only part of the HSC niche but are also required for HSC maintenance and hematopoiesis [52,53]. Based on this, we developed a novel and disruptive technology to isolate human immature dental pulp stem cells (hIDPSCs)—a special type of MSC that expresses high levels of nestin [19]. For this reason, we analyzed here the therapeutic potential of hIDPSCs, produced on c-GMP, for the treatment of acquired aplastic anemia (AA), using a mouse model subjected to the total body irradiation (TBI) model to study the acquired AA.

In this sense, we provided first-time evidence of significant mouse BM content improvement observed in the AA model following three consecutive hIDPSC (1 × 10^6^ cells/animal) transplantations. Furthermore, we observed that after three consecutive hIDPSC transplantations (1 × 10^6^ cells/animal), the irradiated mice showed high BM cellularity, recovering the normal BM histology after 62 days of cell transplantation (short-term treatment), when compared to irradiated mice treated with saline (placebo). For the first time, we showed that at D182 the irradiated and the hIDPSC-treated mice demonstrate stable BM tissue improvement, as evidenced by histological studies, while the irradiated and placebo-treated BM still presented a significant fat deposit in the BM. These data suggest that hIDPSCs can stimulate BM tissue recovery and long-term hematoprotection. Overall, these results were obtained by analyzing the short-term MSC treatment effect. It is not clear whether this effect would be stable for a long time. Generally, a single MSC transplant was used, and the doses ranged from 1 × 10^6^ to 2.5 × 10^7^ cells per mouse, using intravenous or intraperitoneal (IP) routes [37,38,54,55,56]. To confirm this, we demonstrated evidence that the hIDPSCs successfully grafted within the bone marrow (BM) and spleen of mice subjected to TBI. The cell engraftment was demonstrated by the presence of anti-hNu-positive cells in the BM and spleen. A few studies demonstrate MSCs homing into BM [57]; hIDPSC homing was still observed in mouse BM and spleen 30 days after irradiation and hIDPSC treatment. These data not only demonstrate the cell engraftment, but also suggest that these cells can support the hematopoiesis. 

Supporting this evidence, we also showed an increased number of colony-forming units (CFU) of BM cells in an irradiated mouse group treated with hIDPSC, when compared to the irradiated mouse group treated with saline. This result suggests that the hIDPSCs (which have an ectomesenchymal origin from the neural crest) offer support to residual HSCs, facilitating the recovery of BM after the TBI, as verified with typical MSCs [10,58,59].

We demonstrated a significant reduction in endogenous nestin+ and CD44+ in BM in irradiated mice. However, we observed an enrichment of nestin+ cells at day 62 in the irradiated and hIDPSC-treated mice. Nestin is a selective marker of BM-MSC, and rare stromal Nestin+ cells are known to coordinate HSC traffic under homeostasis [21]. They also helped support the BM “niche” formed by MSCs and HSCs. The endogenous CD44+ cell increase in mouse BM was observed only 180 days after hIDPSC transplantation. CD44 is essential for human hematopoietic regulation, including lymphocyte migration and activation, progenitor cell proliferation, and BM environment restoration. The involvement of CD44 in HSC homing and niche embedding is known and has been used as a means to mobilize HSC. BM stroma formation also requires CD44 that supports the process by induction of IL-6 secretion [60,61].

The analysis demonstrates that the hematimetric parameters at D2 (just after irradiation) decrease significantly, and they are still low at D17, after the first hIDPSC transplantation. However, they increase after the second hIDPSC transplantation (D32) and maintained stability until the end of the experiment (D182). Accordingly, Diaz et al., (2020) [40] showed that leukocytes were rapidly lost within 3 days after irradiation, while the red blood cells remained at 17 days. Interestingly, regarding the MSC capacity to rescue hematopoiesis, different studies provide conflicting results. Thus Kim et al. (2018) [62] report that umbilical cord MSCs can rescue hematopoiesis after three hours of radiation exposure. In contrast, Diaz et al. (2020) [40] demonstrated that BM MSCs were not able to restore endogenous hematopoiesis after radiation, while reproducing exactly the protocol of Kim et al. (2018) [62]. Other authors communicate that murine MSC could improve survival when administered alone within 24 h of radiation [55] however, the study failed to evaluate the changes in hematologic parameters in the peripheral blood or bone marrow. Despite this effect, we observed that only the value of the RBCs reached normal erythrocyte levels (>11 × 10^6^/μL) after the second hIDPSC transplantation (D32) and maintained stability until the end of the experiment (D182). This result suggests that MSCs may contribute to the differentiation of the HSC to the myeloid progenitor cell, which is responsible for the formation of RBC. Supporting this hypothesis, we observed a presence of some myeloid progenitor as well as megakaryocytes within the BM of the irradiated mouse group treated with hIDPSCs. In this sense, studies have shown that interleukin (IL)-6 plays a vital role in hematopoiesis, inducing the megakaryocyte development and platelet formation [63,64]. Thus, considering that MSCs express and secrete high levels of interleukin (IL)-6 [65,66], we suggest that this may contribute to the megakaryocyte formation, increasing the number of RBCs. Supporting this evidence, Diaz et al. [40] showed that bone marrow stromal cell therapy improves mouse survival after the exposure to ionizing radiation but does not restore endogenous hematopoiesis. Similar data were reported by Yang et al., 2021 [67], which provided evidence that human umbilical cord MSC transplantation restored and promoted the recovery of radiation-induced hematopoietic damage in a mouse model. 

Despite this evidence, novel studies are necessary to better understand the role of MSCs in hematopoiesis.

## 5. Conclusions

Our data demonstrate that although hIDPSC differs from BM-MSC and other typical MSCs (derived from the other mesenchymal tissues), they can restore the normal BM histology and its function in the AA animal model. These data make the hIDPSCs a useful and alternative source of therapeutic cells for the treatment of acquired AA. However, only the clinical investigation could provide evidence for the hIDPSC usefulness for human AA.

## Figures and Tables

**Figure 1 cells-11-02252-f001:**
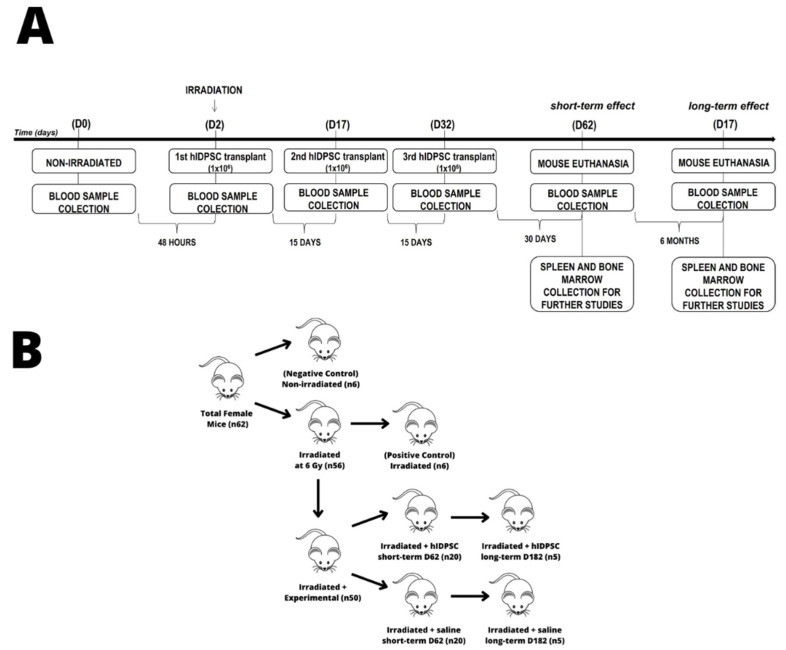
Flowchart showing the cellular therapy with hIDPSCs used in this study: (**A**) On day 0 (D0, pre-irradiation), peripheral blood was collected from all mice as a control. After 48 h of TBI exposure (day 2, D2), the mice received the first cell (1 × 10^6^ hIDPSC/animal) or saline (100 μL) administration (by intraperitoneal via). The second cell or saline administration was performed 15 days after the first administration (day 17, D17), and the third cell or saline administration, 15 days after the second administration (day 32, D32). Three days after the last cell or saline administration (day 62, D62), 40 mice (20 receiving cells and 20 receiving saline) were euthanized to analyze the short-term efficacy of the therapy. To evaluate the long-term efficacy of the therapy, 10 mice (5 receiving cells and 5 receiving saline) were euthanized six months after the last cell or saline administration (day 182, D182). Blood samples were collected on D2, D17, D32, D62, and D162. On D32 and D182 (euthanasia), the spleen was collected for analysis. Flowchart showing the groups used in this study: (**B**) In this study 62 isogenic C57BL/6 female mice (n = 62), aged 4-6 weeks were used. Six non-irradiated mice (n = 6) were used; they were employed as a negative control. A total of 56 mice (n = 56) were irradiated with TBI at a radiation dose of 6Gy. As a control, 6 mice (n = 6) were used from the irradiated group of 56; they were euthanized 48 h after TBI radiation to confirm the BM ablation (positive control). The other 50 mice (n = 50) subjected to TBI exposure were divided into two groups. The first group (n = 25) was intraperitoneally treated with three doses of 1 × 10^6^ hIDPSC/animal. The second group (n = 25) was intraperitoneally treated with three doses of 100 μL of saline (0.9% NaCl solution, placebo). Forty mice (n = 40) were euthanized 30 days after third cell (n = 20) or saline (n = 20) administration (short-term/D62), while ten animals (n = 10) were euthanized only 6 months after third cell (n = 5) or saline (n = 5) administration (long-term/D182).

**Figure 2 cells-11-02252-f002:**
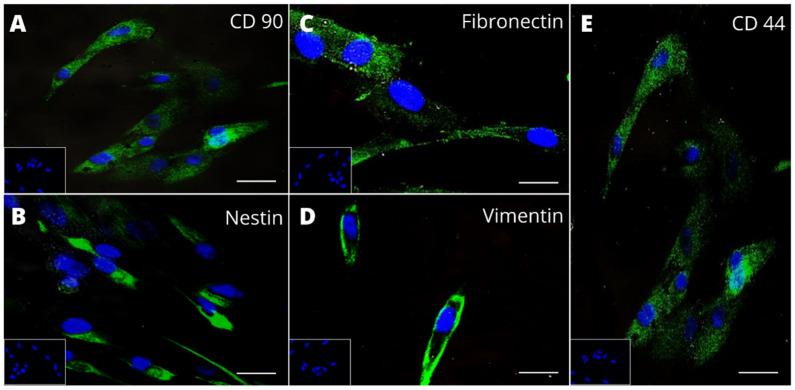
Analysis of CD90, fibronectin, nestin, vimentin, and CD44 expression in undifferentiated hIDPSCs. Epifluorescence shows positive immunostaining (FITC) for CD90 (**A**), nestin (**B**), fibronectin (**C**), vimentin (**D**), and CD44 (**E**), respectively. Nucleus is stained with DAPI (blue). Scale bars = 50 µm (40×).

**Figure 3 cells-11-02252-f003:**
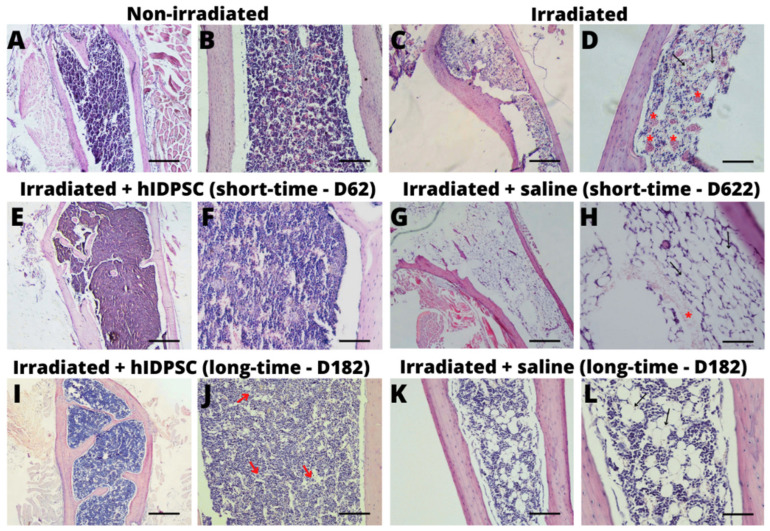
Microphotography of BM sections stained with hematoxylin and eosin (HE) at a magnification of 4× to 20×. Subfigures (**A**) (4×) and (**B**) (20×) show spinal canal filled with hematopoietic elements in BM without irradiation. Subfigures (**C**) (4×) and (**D**) (20×) show a hypoplasic BM, with decreased cellularity, a greater region of the spinal stromal (indicated by the arrows) and infiltrated red blood cells (indicated by the larger asterisk in red) in BM of the irradiated group (48 h after irradiation). Results also show a fat replacement as a consequence of the BM failure. Subfigures (**E**) (4×) and (**F**) (20×) show BM of hIDPSC-treated group (short-term, 62 days), demonstrating the recovery of BM, which was evidenced by the high cellularity. Similar results were observed in BM of hIDPSC-treated group (long-term, 182 days) (**I**,**J**). Subfigures (**G**) (4×) and (**H**) (40×) show a hypoplasic BM, with decreased cellularity in irradiated mice treated with saline (short-term, 62 days). Similar results were observed in BM of saline group (long-term, 182 days) (**K**,**L**), suggesting that residual HSCs present within the BM after the TBI are not able to recover the BM of mice. Scale bars = 500 µm (4×) and 100 µm (20×).

**Figure 4 cells-11-02252-f004:**
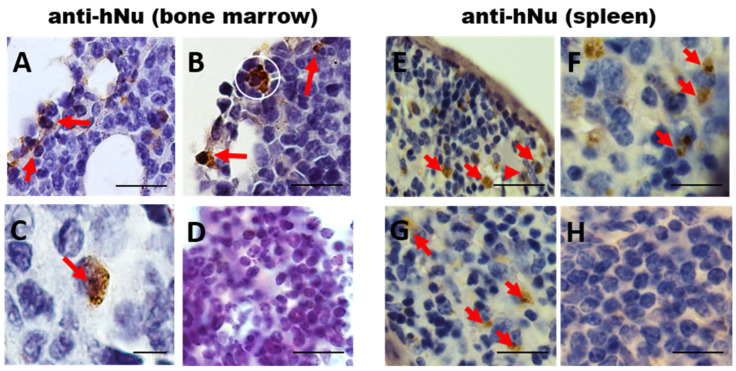
Photomicrography showing the presence of anti-hNu-positive cells (hIDPSCs) within the bone marrow (**A**–**C**) and spleen (**E**–**G**)—red arrow, confirming that the hIDPSCs are engrafted into these sites. Results also show the absence of unspecific labelling of the secondary antibody in the untreated group and (**D**,**H**), confirming the specificity of the anti-human nucleus antibody. Scale bars = 100 µm (20×), 50 µm (40×), 10 µm (100×).

**Figure 5 cells-11-02252-f005:**
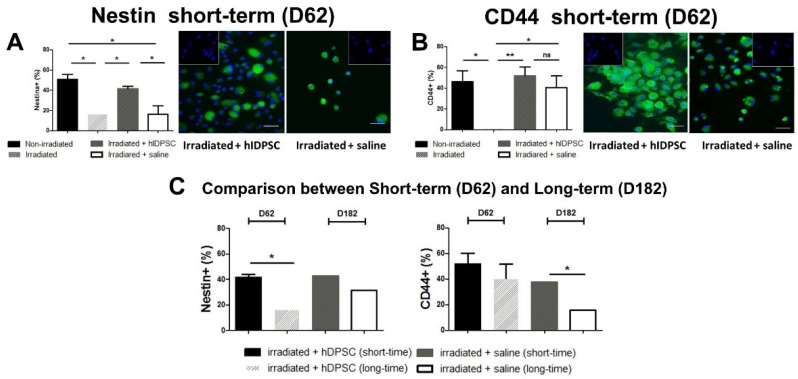
Results of nestin and CD44 immunodetection in bone marrow of different irradiated mouse groups by flow cytometry and immunofluorescence in the short term (D62) (**A**,**B**) and comparison of nestin and CD44 immunodetection in bone marrow between groups treated with hIDPSC and saline in the short term (D62) and long term (D182), respectively. Results show that the exposure to TBI reduced the number of nestin- (**A**) and CD44-positive cells (**B**) 48 h after the irradiation in relation to non-irradiated mice (*). However, the treatment with hIDPSCs increased the number of nestin- (*) (**A**) and CD44-positive cells (**) (**B**) in relation to the irradiated. These results were also confirmed by immunofluorescence (**A**,**B**). Results show statistical difference in the number of nestin-positive cells between the hIDPSC-treated group and those treated with saline (*) (**A**). In contrast, the saline group showed a statistical increase (*) in the number of CD44-positive in relation to the irradiated (**B**). The graph (**C**) is the comparison between irradiated + hIDPSC-treated group and irradiated + saline group in the short term (D62) and irradiated + hIDPSC-treated group and irradiated + saline group in the long term (D182). The results showed the statistical difference between the group treated with hIDPSC and saline (*) in percentage of nestin-positive in the short term (D62) (**A**) and after 6 months or D182 (long-term) did not show a statistical difference (**C**). In addition, the percentage of CD44-positive did not show a statistical difference between group treated with hIDPSC and saline in the short term (D62) (**B**), while only after 6 months or D182 (long-term) showed the statistical difference between hIDPSC-treated and saline groups (*) (**C**). Statistical analysis performed through ANOVA, followed by Tukey post hoc test (both with significance level of 5%). *p*-values < 0.001 (**), or < 0.05 (*) indicate significant differences. *p*-values > 0.05 indicate absence of significant differences (ns—non-significant). Scale bars = 50 µm (40×).

**Figure 6 cells-11-02252-f006:**
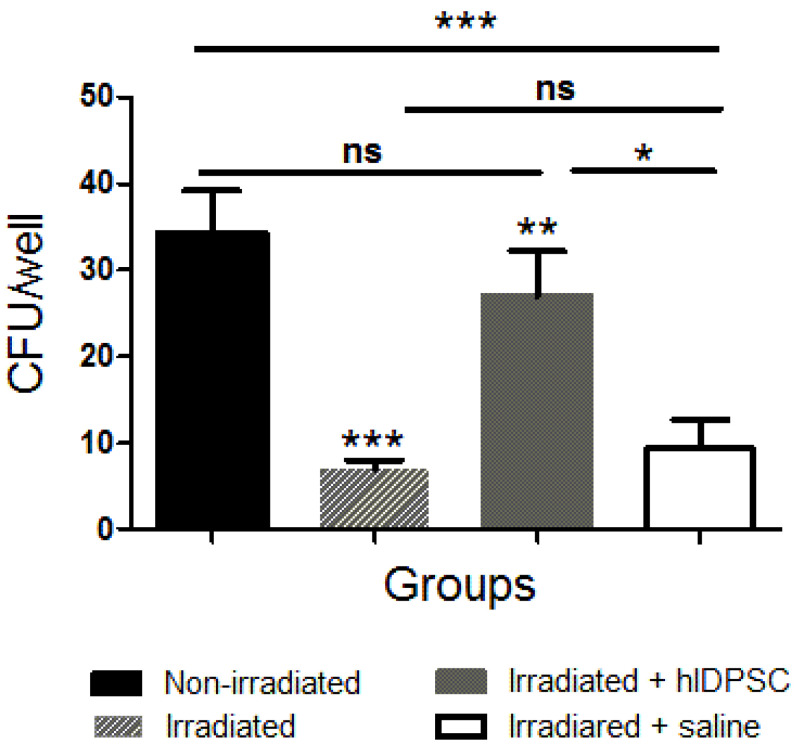
Comparison of the number of colony-forming units (CFUs) among different mouse groups. Results show that the number of CFUs was significantly reduced 48 h after irradiation (***) and remains the saline group in comparison to the non-irradiated (***). However, it is verified that the treatment with hIDPSCs increased the number of CFUs (**) to values statistically similar to the non-irradiated mouse group. In addition, there is a statistical difference between the hIDPSC-treated group and the saline group (*). Statistical analysis performed through ANOVA, followed by Tukey post hoc test (both with significance level of 5%). *p*-values < 0.0001 (***), < 0.001 (**), or < 0.05 (*) indicate significant differences. *p*-values > 0.05 indicate absence of significant differences (ns—non-significant).

**Figure 7 cells-11-02252-f007:**
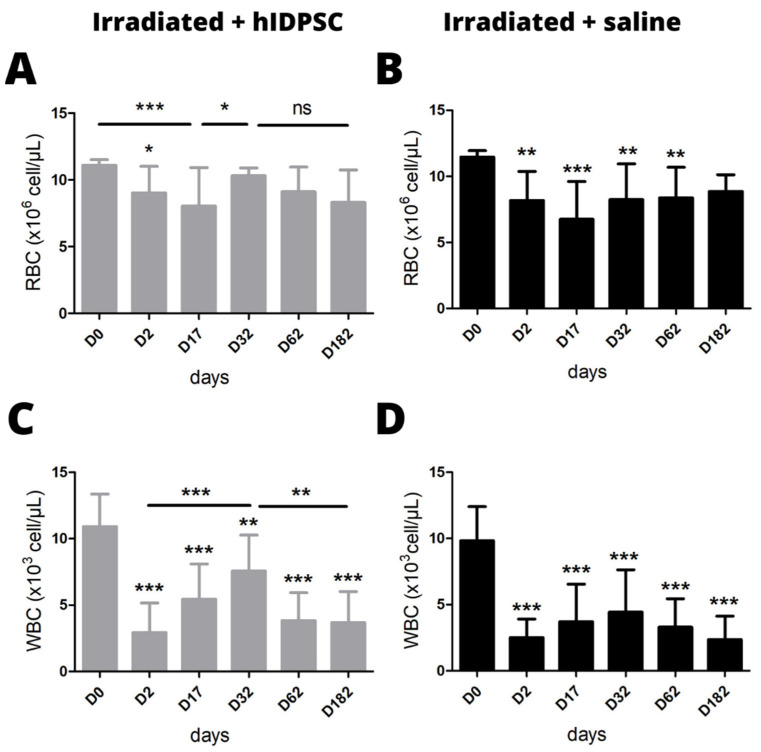
Evaluation of erythrocyte values (×10^6^/μL) (**A**,**B**) and leukocyte values (×10^3^/μL) (**C**,**D**) of the group treated with hIDPSCs (**A**,**C**) or non-treatment (**B**,**D**) as a function of time (days). Before irradiation (D0); 48 h after irradiation (D2); 15 days after the first transplant (D17); 15 days after the second transplant (D32); and 30 days after the third transplant (D62). (**A**) The group treated with hIDPSCs displays a significant decrease on D2 (*) and D17 (***) but a significant recovery (*) on D32, reaching normal erythrocyte levels (>11 *×* 10^6^/μL) up to D62 until D182. (**B**) The saline group displays consecutive drops in erythrocyte levels up on D2 (**), D17 (***), D32 (**) to the time of euthanasia on D62 (**). (**C**) The group treated with hIDPSCs displays a significant drop of the absolute values of leukocytes on D2 (***), D17, (***), D32 (**), D62 (***), and D182 (***). However, there is a significant recovery in leukocyte values (***) after the second transplant (D32). (**D**) The non-treatment group displays a significant drop after irradiation (***), which was maintained until the end of the study (D182). Asterisk indicates statistical significance 18 (ANOVA) in the experimental groups, *p* < 0.05 (*), *p* < 0.01 (**) and *p* < 0.001 (***). Graphs are generated by GraphPad Prism 5 software (GraphPad; PRISM, 2007).

**Table 1 cells-11-02252-t001:** Primary antibodies employed to perform FC—flow cytometry; IHC-P—immunohistochemistry of paraffin embed tissue, IF—immunofluorescence methodologies.

Antibody	Species	Brand	Dilution
CD44	Rat	AbCam, Cambridge, UK (Ab40983)	1:100
CD90	Rat	AbCam, Cambridge, UK (Ab3105)	1:100
Fibronectin	Rabbit	Dako, California, USA (0245)	1:200
Nestin	Rabbit	AbCam, Cambridge, UK (Ab105389)	1:200
Human Nucleus	Mouse	AbCam, Cambridge, UK (Ab191181)	1:500
Vimentin	Goat	Santa Cruz Biotechnology, Texas, USA (sc7557)	1:100

**Table 2 cells-11-02252-t002:** Second antibodies employed to perform FC—flow cytometry; IHC-P—immunohistochemistry of paraffin embed tissue, IF—immunofluorescence methodologies.

Antibody	Species	Brand	Dilution
Anti-Goat IgG (H+L) (Alexa Fluor^®^ 647 conjugate)	Goat	Thermo Fischer Scientific, California, USA (A21236)	1:2000
Anti-Goat IgG (FITC conjugate)	Goat	Santa Cruz Biotechnology, TX, USA (sc2079)	1:1000
Anti-Goat IgG (HRP conjugate)	Goat	Dako, California, USA (p0449)	1:100
Anti-Rabbit IgG (H+L) (Alexa Fluor^®^ 633 conjugate)	Rabbit	Thermo Fischer Scientific, California, USA (A21070)	1:2000
Anti-Rabbit IgG (FITC conjugate)	Rabbit	Santa Cruz Biotechnology, TX, USA (sc2012)	1:1000
Anti-Mouse/Anti-Rabbit Envision+ Dual Link (HRP conjugate)	Mouse/Rabbit	Dako, California, USA (k4063)	-
Anti-Rat IgG (FITC conjugate)	Rat	Santa Cruz Biotechnology, Texas, USA (sc2011)	1:1000
Anti-Rat IgG (HRP conjugate)	Rat	Dako, California, USA (P0450)	1:100

## Data Availability

Not applicable.

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
