# Peer review of "Therapeutic Potential of Human Immature Dental Pulp Stem Cells Observed in Mouse Model for Acquired Aplastic Anemia"

_cells, 2022, doi:10.3390/cells11142252_

Round 1

Reviewer 1 Report

The Authors have considerably improved the description of experimental design in Materials and Methods. However, many important concerns are still present:

- the authors answered to my previous revision by writing that all the antibodies used in the study were specific for mouse, (except human anti nuclei). However, in Fig.2, they showed the characterization of hiDPSC which resulted positive for nestin and CD44...are these the same antibodies used later to demonstrate the "endogenous" marker presence? are they different?Moreover, and more important:  they used an antibody anti human nuclei produced in mouse, as stated in Table 1, in a mouse model...the secondary antiblody may recognize every murine protein...it is not a good choice!

-Concerning cellularity: the authors evidenced just megakaryocytes, why? other cells?

-CFU assay: it is really unclear when the assay has been performed...it seems it has performed 48hours after irradiation...but when the hiDPSC were administrated? at D2? that is at the same time? how could you see a difference, an effect of these cells?

-concerning nestin and CD44: if the same antibody recognizes hiDPSC and endogenous cells…you can’t discriminate and you can’t affirm which cells express them. Fig. 3: There is a discrepancy in nestin amount in panel A and C short term. More in details nestin in panel C is at the same amount in irradiated and transplanted and irradiated and saline…

 And about CD44…it is increased after 6 months with respect to saline group…but all these analysis has been performed on just 5 animals…

More in general: I think that the most important result is the demonstration that also stem cells from dental origin could contribute to bone marrow repopulation. To give a further significance to the work, the results obtained with hiDPSC should be compared to those obtained with stem cells used a gold standard in such a regenerative therapy, in order to clarify the real therapeutic potential of hiDPSC.

Author Response

Reviewer 1

We thank the Reviewer for constructive critics of our work that help to improve their quality.

Comment: the authors answered to my previous revision by writing that all the antibodies used in the study were specific for mouse, (except human anti nuclei). However, in Fig.2, they showed the characterization of hiDPSC which resulted positive for nestin and CD44...are these the same antibodies used later to demonstrate the "endogenous" marker presence? are they different?

Response: We thank the Reviewer for constructive critics of our work that help to improve their quality. We inform that the answers were reviewed and we apologize for misunderstanding. Both CD44 and Nestin markers are the same ones used in the characterization of hIDPSC cells as well as to demonstrate endogenous labeling of mouse bone marrow. The antibodies are reactive with human and mouse material.

Comment: -concerning nestin and CD44: if the same antibody recognizes hiDPSC and endogenous cells…you can’t discriminate and you can’t affirm which cells express them. 

Response: We apologize for this misinterpretation. It is important to note that as the markers are not specific only for mice or humans, we cannot affirm that the presence of Nestin positive or CD44 positive is due to the presence of hIDPSC in the tissue, since these markers may be present in the murine bone marrow. As a result, we only reported an increase in expression in the treated group after irradiation, but we do not know if these positive cells are from hIDPSC grafted at the bone marrow or if it stimulated endogenous production in the murine bone marrow cells. Furthermore, we are aware that the placebo group was able to express the same markers and that they may also have a physiological recovery. Therefore, we always affirm that the treatment with our product is a support and that we observed that the animals who received the treatment in general had a faster response to improvement than the placebo animals.

Comment: Fig. 3: There is a discrepancy in nestin amount in panel A and C short term. More in details nestin in panel C is at the same amount in irradiated and transplanted and irradiated and saline…

Response: We thank for this comment, but the figure 3 is the microphotography of BM sections. We think the comment were about the figure 5. For this reason, we are answering for the question about figure 5. If this is a mistake, please forgive us for the misunderstanding. The graph (C) is the comparison between irradiated + hIDPSC treated group and irradiated + saline group in short-term (D62) and irradiated + hIDPSC treated group and irradiated + saline group in long-term (D182). While graph A shows the percentage of nestin positive cells in the groups: non-irradiated, irradiated, irradiated + hIDPSC treated and irradiated + saline groups (only in short-term - D62). The long-term group was an additional evaluated after 6 months and that is why the comparison with the short-term was performed in a separate graph. Therefore, the panel A and C in short-term are at the same amount.

Comment: And about CD44…it is increased after 6 months with respect to saline group…but all these analyses has been performed on just 5 animals…

Response: We thank for this important observation. The long-term group was determined as an additional to the study to evaluate the animals 6 months after hIDPSC therapy. Precisely to verify if the animals would recover physiologically (saline group) the parameters evaluated. We chose to study this additional group to be sure of the therapy proposal, although with a smaller number of animals in this group. Due to the total number approved by the ethics committee. We rearranged the long-term group with a minimum number capable of generating statistical results.

Comment: -Moreover, and more important:  they used an antibody anti human nuclei produced in mouse, as stated in Table 1, in a mouse model...the secondary antiblody may recognize every murine protein...it is not a good choice!

Response: We thanks for this comment and advice. The antibody anti-human was produced in mouse. We tried to choose another option, but normally antibodies for detecting human cells are produced in mouse. Therefore, we showed that the group that did not receive hIDPSC (human cell) did not present any presence of labeling in the immunohistochemical reaction. With this, we can observe that there was no unspecific labeling. We made the immunohistochemistry reaction very carefully throughout the process, especially to avoid unspecific connections, blocked with 5% BSA solution (in PBS) for 40 minutes.

Comment: -Concerning cellularity: the authors evidenced just megakaryocytes, why? other cells?

Response: We thank for this comment. In order to clarify, as requested, we rewritten the passages that was only mentioned megakaryocytes. The histology and medullary smear showed the general medullary tissue recover, including the presence of several precursors as showed below:

“BM destruction was evidenced by the: disseminated lesion along with the medullary tissue and reduction of cellularity (hypoplasia/medullary aplasia), absence of medullary precursors cells in general, infiltration of red blood cells, more significant amount of spinal stromal, and substitution of BM by fat elements in some regions (Figure 3 D).”

“We identified the presence of the spinal canal filled with medullary precursors cells in general, like megakaryocytes into the BM of hIDPSC-transplanted mice (Figure Supplementary 1).”

Comment: - CFU assay: it is really unclear when the assay has been performed...it seems it has performed 48hours after irradiation...but when the hiDPSC were administrated? at D2? that is at the same time? how could you see a difference, an effect of these cells?

Response: We thank for this answer. We evaluated the difference and the effect of bone marrow through the CFU methodology comparing the bone marrow of each group. We need to euthanize the animal to collect the bone marrow medullar aspirate to perform the CFU assay. With this, we organize different groups, with different euthanasia points. The CFU assay was performed in the non-irradiated group (control without irradiation and euthanized on D0), irradiated group (control irradiated and euthanized 48 hours after irradiation – D2) and in the experimental groups: irradiated + hIDPSC treated and irradiated + saline. The experimental groups were irradiated (D0) and 48 hours after irradiation (D2) they started the treatments consisting of three hIDPSC doses or saline every 15 days. And only 30 days after the third administration they were euthanized (D62).

Comment: More in general: I think that the most important result is the demonstration that also stem cells from dental origin could contribute to bone marrow repopulation. To give a further significance to the work, the results obtained with hiDPSC should be compared to those obtained with stem cells used a gold standard in such a regenerative therapy, in order to clarify the real therapeutic potential of hiDPSC.

Response: We very thank for the considerations and agree with the points raised. Also, we tried to use others mesenchymal stem cells therapy studies with different sources to compare to our study. And we observed that like other studies an improvement after therapy.  -          HU, et al. (2010). The radiation protection and therapy effects of mesenchymal stem cells in mice with acute radiation injury.  Furthermore, we showed a repopulation of the bone marrow and we also brought additional methodologies as well as evaluate the animals 6 months after the treatment, which other studies were unable to demonstrate. Therefore, we reviewed the manuscript and followed the reviewer's consideration to clarify the real therapeutic potential of hIDPSC.

We thank for this comment. The discussion was rewritten in order to provide the explanations required.

Reviewer 2 Report

Revisions accepted

Author Response

We thank the Reviewer for constructive critics of our work that help to improve their quality.

Reviewer 2

Comment: Revisions accepted

Response:  We thank the Reviewer for this comment and for accepted all the changes made as requested previously.

Reviewer 3 Report

In the paper by Gonzaga et al. the authors show the potential of dental pulp stem cells for recovery of the hematopoietic system after irradiation and suggest that these cells could be a potential therapeutic tool for aplastic anaemia.

I have now major concerns on the experiments or the methodology, which seem solid, the number of mice seems adequate and authors  have done colony forming assays to show recovery of HSCs and analysed the blood cell counts in he mice. While the tissue sections (figure 3) show clear recovery in the cell treated animals and the CFU assay support therapeutic potential of the cells, some of the data is very confusing and hard to understand. And the data in figures 5 and 7 does not seem to fully support the conclusions.

First of all I cannot understand the Nestin and CD44 results in figure 5. The results in figure 5C do not seem to match those in figures 5A and B. Further,  also saline treatment seems to increase CD44 positivity, suggesting that this is not due to the cell transplantation. To make this even more confusing, the figure shows an asterisk in nestin expression between the cell and saline treated mice, whereas the legend says that there is no statistical significance. 

Secondly, the blood cell count data in figure 7 is hard to understand. If day 0 is before irradiation and d2 after irradiation, shouldn’t all other time points be compared to D2, now it is impossible to tell what is significant and what not, when the comparisons are done as they are. In the discussion they state that RBC count is high at D62, however, looking at the data, it seems like there is no clear difference to D2 at this time. 

The authors should clarify these aspects and make sure the the conclusion are supported by the data before the paper can be accepted in Cells. 

Minor comments:

Figure 1 shows irradiation at day2, shouldn’t this be at day 0?

Figure 1 legend row 152 third days? should this be thirty days. Lots of repetition in figure legend 

Row 97 cultivated until twice

Row 120 typo minutes

Row 178 typo Table 2

Row 189  typos described in figure 1

Row 206 typo room temperature

Author Response

We thank the Reviewer for constructive critics of our work that help to improve their quality.

Reviewer 3

Comment: While the tissue sections (figure 3) show clear recovery in the cell treated animals and the CFU assay support therapeutic potential of the cells, some of the data is very confusing and hard to understand. And the data in figures 5 and 7 does not seem to fully support the conclusions.

Response: We thank the Reviewer for constructive critics of our work that help to improve their quality. We apologize for this misinterpretation and reviewed the conclusion and rewritten, as showed below:

“Our data demonstrate that although hIDPSC differs from BM-MSC and other typical MCS (derived from the other mesenchymal tissues), they can restore the normal BM histology and and its function in the AA animal model. These data make the hIDPSCs a useful and alternative source of therapeutic cell for the treatment of acquired AA. However, only the clinical investigation could provide evidence for hIDPSC usefulness for human AA.”

And we decided to take more care in asserting the potential of hIDPSC therapy. We mostly concluded that hIDPSC treatment is able to repopulate the mice bone marrow after the irradiation process, which mimics the aplastic anemia animal model. We would like to reinforce that this is the main conclusion of the work with the CFU proof. However, the evaluation of the hematological elements through the hemogram and the assays of expression of endogenous markers present in the bone marrow, are additional assays in the study to verify the potential of hIDPSC therapy. With this, we would like to reinforce that the repopulation of the medullary environment after hIDPSC transplantation is the main support for the conclusion.

Comment: First of all I cannot understand the Nestin and CD44 results in figure 5. The results in figure 5C do not seem to match those in figures 5A and B.

Response: The graph (A and B) shows the percentage of nestin positive cells in the groups: non-irradiated, irradiated, irradiated + hIDPSC treated and irradiated + saline groups (only in short-term - D62) of nestin and CD44, respectively. The graph (C) is the comparison between irradiated + hIDPSC treated group and irradiated + saline group in short-term (D62) and irradiated + hIDPSC treated group and irradiated + saline group in long-term (D182) of nestin and CD44, respectively. The long-term group was an additional evaluated after 6 months and that is why the comparison with the short-term was performed in a separate graph.

Comment: Further, also saline treatment seems to increase CD44 positivity, suggesting that this is not due to the cell transplantation.

Response: We thank for the comments and apologize for the misinterpretation. We are aware that the placebo group was able to express the same markers and that they may also have a physiological recovery. Therefore, we always affirm that the treatment with our product is a support and that we observed that the animals who received the treatment in general had a faster response to improvement than the placebo animals. We reported as the result, an increase in expression in the treated group after irradiation, but we do not know if these positive cells are from hIDPSC grafted at the bone marrow or if it stimulated endogenous production in the murine bone marrow cells.

Comment: To make this even more confusing, the figure shows an asterisk in nestin expression between the cell and saline treated mice, whereas the legend says that there is no statistical significance. 

Response: We apologize for this misinterpretation. For this reason, the legend of figure 5 were modified, providing details about the statistical significance as showed below:

“Results show statistical difference in the number of nestin-positive cells between the hIDPSC treated group and those treated with saline (*) (A).”

Comment: Secondly, the blood cell count data in figure 7 is hard to understand. If day 0 is before irradiation and d2 after irradiation, shouldn’t all other time points be compared to D2, now it is impossible to tell what is significant and what not, when the comparisons are done as they are.

Response: We thank for the comments and apologize for the figure misinterpretation. We performed the statistcial analyses using the one-way analysis of variance (ANOVA), followed by the Tukey test, both with a significant level of 5% using the GraphPad Prism 5.02 software. Then. all the point are compared with each other. Therefore, in order to facilitate the interpretation, the Figure 7 was corrected.

Comment: In the discussion they state that RBC count is high at D62, however, looking at the data, it seems like there is no clear difference to D2 at this time. 

Response: We apologize for this misinterpretation and rewritten, as showed below:

“After the significant increasement presented on D32, the values of RCB did not show statistical decrease until D182 (Figure 7).”     

We thank for this comment. The discussion was rewritten in order to provide the explanations required.

Round 2

Reviewer 1 Report

The paper is acceptable for publication

This manuscript is a resubmission of an earlier submission. The following is a list of the peer review reports and author responses from that submission.

Round 1

Reviewer 1 Report

In the current manuscript, the authors study the therapeutic plasticity and protective role of hIDPSC for Aplastic anemia. The scientific design is well planned and supported by sufficient experiments. However, the manuscript needs extensive language editing. There are multiple grammatical and spelling mistakes.

One major concern in this study, is the use of banked dental pulp stem cells from the year 2006/2011. The authors did not re examine these cells for changes and relied on previous characterization. It would have been better to use freshly isolated cells or at least subject these cells for quality control check to make sure there are no changes in the used population.

Also, section 2.4 regarding cell therapy needs to be rewritten  with more details and explanation of doses to avoid confusion.

.

Reviewer 2 Report

The paper by Gonzaga et coll. investigated the potential of dental pulp stem cells as therapeutic tool to treat aplastic anemia. Bone marrow derived Mesenchymal stem cells have been already investigated for that aim, but dental pulp stem cells could be simpler available.

In general, I found a little confusing some descriptions in material and methods :

  • In "Cell therapy" paragraph which is the difference between the two groups? what is the meaning for the administration of 1 ml of cell suspension, without any information about the concentration?
  • In "Cell therapy" paragraph the forty mice sacrified after 30 days...from which group?

In Results:

  • why nestin? it is a marker of immature neural cells...the explanation for the choice of the markers is too vague
  • all the part concerning the short and the long term benefits of hIDPSC is based on the images, they are valid, but they are not quantitative, a quantitative analysis is necessary
  • about the expression of endogenous markers: are all your antibodies specifically designed to discrimitae between human and murine markers? how could you affirm that the effect was on irradiated stroma?
  • Fig. 5 the graphs have not the same y-axis scale
  • what is the meaning of irradiation+ non treatment group? and why in CD44 this group showed an improvement?
  • BM-colony forming unit assay: how could you affirm that the colony forming cells you observed were the residual cells of bone marrow? again, what is the meaning of the "non-treatment" group of Fig. 6?
  • Fig.7: what is the reason of the geenral improvement observed at D32?  why C and D have different y-axis scale?
  • discussion is very little convincing and it did not offer an explanation for the differences observed between CD44...morover, you affirm that nestin is a selective marker of BM.MSC, but it is not correct, and this marker is present also in your negative control